# Effect of Prenatal Opioid Exposure on the Human Placental Methylome

**DOI:** 10.3390/biomedicines10051150

**Published:** 2022-05-17

**Authors:** Kristyn N. Borrelli, Elisha M. Wachman, Jacob A. Beierle, Elizabeth S. Taglauer, Mayuri Jain, Camron D. Bryant, Huiping Zhang

**Affiliations:** 1Laboratory of Addiction Genetics, Department of Pharmacology and Experimental Therapeutics, Boston University School of Medicine, Boston, MA 02118, USA; kristynb@bu.edu (K.N.B.); jbeierle@bu.edu (J.A.B.); camron@bu.edu (C.D.B.); 2Graduate Program for Neuroscience, Boston University, Boston, MA 02118, USA; 3Transformative Training Program in Addiction Science, Boston University, Boston, MA 02118, USA; 4NIGMS Biomolecular Pharmacology Ph.D. Training Program, Boston University School of Medicine, Boston, MA 02118, USA; 5Department of Pediatrics, Boston Medical Center, Boston, MA 02118, USA; elisha.wachman@bmc.org (E.M.W.); elizabeth.taglauer@bmc.org (E.S.T.); 6Boston University School of Public Health, Boston University, Boston, MA 02118, USA; jainm@bu.edu; 7Department of Psychiatry, Boston University School of Medicine, Boston, MA 02118, USA; 8Department of Medicine (Biomedical Genetics), Boston University School of Medicine, Boston, MA 02118, USA

**Keywords:** prenatal opioid exposure, placenta, DNA methylome, differential methylation, functional annotation, neonatal opioid withdrawal syndrome

## Abstract

Prenatal exposure to addictive drugs can lead to placental epigenetic modifications, but a methylome-wide evaluation of placental DNA methylation changes after prenatal opioid exposure has not yet been performed. Placental tissue samples were collected at delivery from 19 opioid-exposed and 20 unexposed control full-term pregnancies. Placental DNA methylomes were profiled using the Illumina Infinium HumanMethylationEPIC BeadChip. Differentially methylated CpG sites associated with opioid exposure were identified with a linear model using the ‘limma’ R package. To identify differentially methylated regions (DMRs) spanning multiple CpG sites, the ‘DMRcate’ R package was used. The functions of genes mapped by differentially methylated CpG sites and DMRs were further annotated using Enrichr. Differentially methylated CpGs (*n* = 684, unadjusted *p* < 0.005 and |∆β| ≥ 0.05) were mapped to 258 genes (including *PLD1*, *MGAM*, and *ALCS2*). Differentially methylated regions (*n* = 199) were located in 174 genes (including *KCNMA1*). Enrichment analysis of the top differentially methylated CpG sites and regions indicated disrupted epigenetic regulation of genes involved in synaptic structure, chemical synaptic transmission, and nervous system development. Our findings imply that placental epigenetic changes due to prenatal opioid exposure could result in placental dysfunction, leading to abnormal fetal brain development and the symptoms of opioid withdrawal in neonates.

## 1. Introduction

Opioid use disorder (OUD) in pregnancy has increased four-fold over the past decade in the United States (US) and now impacts 1–2% of all pregnancies in the US [1,2]. The primary neonatal outcome, Neonatal Opioid Withdrawal Syndrome (NOWS), typically presents with signs and symptoms of opioid withdrawal after in utero exposure, with often prolonged courses of opioids and lengthened neonatal hospitalizations [3,4]. Prenatal exposure to opioids is known to be associated with adverse pregnancy outcomes, including fetal growth restriction, risk for preterm birth, and higher rates of neonatal intensive care unit (NICU) admission [5,6]. In addition, there is an association between prenatal opioid exposure and risk for altered neuronal function and behavior throughout childhood as well as risk for neurodevelopmental impairment [7,8,9].

The placenta is the critical physiological interface connecting opioid use in pregnant mothers to prenatal exposure of the fetus. In limited cohort studies, maternal OUD has been associated with structural changes within the placental villi, including delayed villous maturation that is associated with stillbirths [10]. Several studies have additionally shown associations between sequestration of toxic substances that inhibit nutrient transport and placental lesions indicating functional insufficiency that includes poor invasion, vascular changes, and vasoconstriction [11,12]. The placenta is also a key target for epigenetic modifications induced by exogenous substances and is the master regulator of the fetal environment [13]. As a uniquely sensitive organ to environmental influences, the placenta is an established site of epigenetic modifications with subsequent differences in gene expression and tissue differentiation [14,15]. Previous studies have shown an association between placental DNA methylation in key genes after exposure to maternal stress and differences in infant neurobehavioral profiles in the first few months after birth [16,17]. These neurobehavioral profiles, as measured by the NICU Network Neurobehavioral Scale (NNNS), are known to be associated with later neurodevelopmental outcomes [16,17]. Though previous studies have demonstrated placental epigenetic modifications induced by exposure to other addictive substances or maternal psychosocial stress, the effect of prenatal opioid exposure in humans on genome-wide epigenomic modifications in the placenta has yet to be explored [17,18,19,20].

The link between maternal OUD and epigenetic modification in other tissues, such as maternal blood and maternal/infant saliva, is well established [21,22]. Opioid exposure in non-pregnant adults is known to lead to changes in DNA methylation in the mu-opioid receptor gene (*OPRM1*; the primary molecular target for the analgesic and addictive properties of opioids), and it has been associated with DNA methylation changes at a genome-wide level [23,24]. In recent years, researchers have examined changes in DNA methylation in *OPRM1* in infants with NOWS as predictive of NOWS severity [25,26]. Specifically, associations between increased cytosine:guanine (CpG) dinucleotide methylation within the *OPRM1* promoter region and NOWS severity have been observed in maternal and infant saliva samples, with increased methylation at select CpG sites associated with more severe NOWS with higher pharmacologic treatment rates, or higher rates of requiring a multi-drug regimen for NOWS treatment [25,26]. Our previous pilot study probed methylation in placental tissues at six CpG sites within the *OPRM1* promoter. Methylation at these sites was not found to be associated with NOWS severity, nor was it associated with opioid exposure [27].

It is well known that there is a high density of hypomethylated CpG sites (or CpG islands) in the promoter region of many genes (including *OPRM1*), and the methylation of promoter CpG sites can inhibit gene transcription. Moreover, altered DNA methylation in the gene body (usually hypermethylated to avoid generating an alternative transcription start site) and the enhancer region (can be up to 1 Mbp away from the gene) can also influence the transcription of genes. Therefore, the present study took a broader and non-biased approach to detect methylome-wide alterations in placental DNA methylation following prenatal exposure to opioids. To our knowledge, genome-wide DNA methylation analyses have not been reported in placental tissues from opioid-dependent women. Identification of genes that contain differentially methylated CpG sites in opioid-exposed placental tissues can facilitate the narrowing of genes associated with NOWS risk and developmental outcomes. Given the associations between epigenetic modifications and differences in infant neurobehavioral profiles following prenatal exposure to drugs of abuse, as well as the findings from our previous studies showing a relationship between NOWS and epigenetic modifications in *OPRM1*, we sought to compare genome-wide DNA methylation in placental tissues of opioid-exposed versus opioid-unexposed pregnancies.

## 2. Materials and Methods

### 2.1. Setting

Boston Medical Center (BMC) is the largest urban safety-net hospital in New England, with a specialized prenatal clinic for women with OUD. Treatment options for pregnant women with OUD include methadone and buprenorphine. BMC practices a rooming-in model of care where infants room-in with their mothers in the postpartum room until maternal discharge, and then the infants are transferred to the pediatric inpatient unit for continued NOWS monitoring and treatment, where their mothers can continue to room-in. The Eat, Sleep, and Console (ESC) method is utilized for NOWS assessments, with an “as needed” methadone pharmacologic treatment protocol.

### 2.2. Subjects

This study was approved by the Boston University Medical Campus Institutional Review Board. Placentas were collected from human subjects between July 2019 and July 2020. Eligibility criteria for inclusion in the opioid cohort included confirmed maternal OUD with opioid exposure for at least 30 days prior to delivery based on chart review and toxicology screening results, singleton pregnancies, gestational age (GA) of ≥36 weeks, and delivery at BMC. For the control group, eligibility criteria for placental collection included delivery at BMC, gestational age (GA) ≥ 36 weeks, and absence of a known substance use disorder per the electronic medical record (EMR) problem list, admission note, and review of toxicology screen lab results. Controls were matched based on month of delivery with the opioid group. Informed consent was waived for this study due to the collection of discarded placental tissue only for these subjects, along with limited de-identified basic demographic data from the EMR. The EMRs for these subjects were accessed once at the time of placental collection, with no identifiers or master code collected per IRB guidelines. As shown in Table 1, 40 pregnant women (20 cases and 20 controls) were recruited for this study. They were all non-Hispanic. Except for two black women (1 case and 1 control), all others were white women (19 cases and 19 controls).

### 2.3. Phenotype Data Collection

Limited data points were collected at the time of placental collection, including maternal race and ethnicity, maternal age, smoking status, gestational age at delivery, infant birth weight, and infant sex. All data were hand abstracted and input into an electronic database. Data were checked for accuracy and missingness, with any discrepancies addressed prior to data analysis.

### 2.4. Experimental Methods

#### 2.4.1. Placental Collection

Placental samples were collected within 2 h of delivery from the Labor and Delivery Unit at BMC by trained research personnel. Placental tissues were taken mid-way between the decidual and chorionic plates, isolating the fetal villous placental tissue (approximately 2–3 pieces and each 0.5–1.0 cm × 0.5–1.0 cm × 0.5–1.0 cm in size). Each sample was placed in a 2 mL cryovial with 1 mL of DNAgard^®^ Tissue solution (Sigma-Aldrich, St. Louis, MO, USA) and stored in a −20 °C freezer in our research laboratory.

#### 2.4.2. DNA Extraction

Genomic DNA was isolated from the placental tissue per standard protocols in batches every 6 months. Genomic DNA extraction was performed by the Boston University Genomics Core lab using the DNeasy Blood and Tissue Kit (QIAGEN, Hilden, Germany), following manufacturer instructions. DNA concentration and purity were quantified using a NanoDrop ND-1000 spectrophotometer (Thermo Fisher Scientific, Waltham, MA, USA), and DNA samples were stored at −20 °C until the Illumina Infinium MethylationEPIC array assay was performed.

#### 2.4.3. Illumina EPIC DNA Methylation Array Assay and Raw Data Processing

Extracted genomic DNA was sent to the Yale Center for Genome Analysis (YCGA) for DNA methylome analysis using the Illumina Infinium MethylationEPIC BeadChip (Illumina, San Diego, CA, USA). The GenomeStudio software (Illumina) was used to generate β values for each CpG site. Data quality control (QC) and normalization, as well as statistical analysis, were performed as in a recent study that investigated peripheral blood DNA methylomic changes in European-American women with OUD using the Illumina Infinium MethylationEPIC BeadChip [23]. Briefly, QC and normalization were performed using the ‘minfi’ R package (v.1.36.0) [28]. Samples were excluded if more than 1% of the probes had a detection *p*-value > 0.01, and probes were excluded if more than half of samples had a detection *p*-value > 0.01. Moreover, cross-reactive probes and probes overlapping genetic variants at targeted CpG sites or single base extension sites, as well as probes with genetic variants overlapping the body of the probe, were excluded [29]. Additionally, probes mapped to X and Y chromosomes were excluded prior to analysis. The ‘ComBat’ method in the ‘sva’ R package (v.3.38.0) [30] was applied to correct for batch effects. Stratified quantile normalization of beta and M values was conducted using the ‘preprocessQuantile’ function in the ‘minfi’ R package (v.1.36.0). This method stratified probes by genomic region (CpG islands, shores, and shelves) to account for differences in region-specific methylation distributions. Density plots were generated to evaluate the distribution of beta (β) values before and after quantile normalization.

#### 2.4.4. Statistical Analysis

All statistical analyses were performed within R v.4.0.3 (www.r-project.org, accessed on 1 March 2022). Baseline demographics for the opioid and control cohorts were summarized. M-values of normalized probe intensity were used for differential methylation analysis, and beta values were used for all data visualization. Differentially methylated probes (DMPs) associated with opioid exposure were identified with a linear model using the ‘limma’ R package (v.3.46.0) [31]. The model included coefficients to account for variance attributable to confounding factors (infant sex, infant birth weight, and batch). In limma, the design matrix was set up as: ~ opioid status + infant sex + infant birth weight + batch. The effect of the ‘opioid status’ coefficient was extracted to determine DMPs. We employed the recommended significance threshold for DNA methylation association analyses of *p* < 9.4 × 10^−8^ as determined from permutation analysis [15]. Methylation of the top differentially methylated CpG sites were visualized via volcano plots [32], heatmaps with hierarchical clustering [33], and site-specific plots of normalized beta values by opioid-exposure group generated with R scripts. To identify differentially methylation regions (DMRs) spanning multiple CpG sites, we utilized the ‘DMRcate’ R package (v.2.4.1) [34].

#### 2.4.5. Bioinformatics Analysis

Gene Ontology (GO) and Kyoto Encyclopedia of Genes and Genomes (KEGG) pathway enrichment analyses of differentially methylated genes were performed using Enrichr [35,36]. Bubble plots of enrichment terms were generated with the R package ‘ggplot2’ v.3.3.3 [37].

## 3. Results

### 3.1. Subjects and Demographic Data

Placental samples (20 control and 20 opioid-exposed) were collected and processed for DNA methylation analysis. One sample from the opioid-exposed group was omitted from the analysis following QC inspection due to a poor signal intensity. Twenty control and 19 opioid-exposed samples were included in the analysis, and demographic information for these samples is shown in Table 1. Opioid-exposed subjects were significantly younger than controls (*p* = 0.010). There were no significant differences in gestational age (GA) at delivery (*p* = 0.220); however, neonates from opioid-exposed moms showed a near-significant trend for lower birthweights compared to those of controls (*p* = 0.050). Notably, five subjects (three opioid-exposed and two controls) had missing data for maternal age, gestational age at delivery, and delivery type.

### 3.2. Differential Methylation in Opioid-Exposed Placentas

Post-filtering, a total of 710,952 CpG sites (from 865,918 CpG sites in total) were used for differential methylation analysis. Density plots of raw and quantile-normalized beta values in opioid-exposed and control groups are shown in Figure 1. There were no individual CpG probes/sites that passed the FDR threshold (*p* < 9.4 × 10^−8^) for significance, with a minimum unadjusted *p*-value of 7.9 × 10^−6^ for cg05771324, which is mapped to the Phospholipase D1 gene (*PLD1*). There were 1,958 CpG sites with an unadjusted *p*-value ≤ 0.005. Of these sites, 684 (~35%) had |∆β| ≥ 0.05 where ∆β = β _(opioid-exposed)_ − β _control_. A volcano plot of ∆β values by −Log_10_
*p*-value (unadjusted) is shown in Figure 2, and the 684 sites with both *p*-value ≤ 0.005 and |∆β| ≥ 0.05 are plotted in red. Of these sites, 397 (~58%) were hypomethylated, and 287 (~42%) were hypermethylated in opioid-exposed samples versus controls. A heatmap displaying normalized beta values indicative of methylation levels at all 684 sites is shown in Figure 3. CpG sites (rows) were clustered by hierarchical sorting, and those with similar trends in methylation were grouped regardless of opioid exposure.

The top 6 CpGs (*p*-value ≤ 0.005 and |∆β| ≥ 0.05), which were mapped to annotated genes, are shown in Figure 4. These sites included cg05771324 (chr3; *PLD1*; 3′UTR; *p* = 7.9 × 10^−6^), cg07585558 (chr6; *RP11-250B2.3*; TSS1500; *p* = 9.4 × 10^−6^), cg22393128 (chr8; *RP11-32K4.1*; 3′UTR; *p* = 2.4 × 10^−5^), cg24415698 (chr13; *AL159152.1*; TSS200; *p* = 3.9 × 10^−5^), cg06547839 (chr7; *MGAM*; TSS1500; *p* = 4.4 × 10^−5^), and cg06347739 (chr11; *ASCL2*; TSS1500; *p* = 5.7 × 10^−5^) (chromosome #; CpG name; gene; genomic region; unadjusted *p*-value). Of these sites, two were hypermethylated in opioid-exposed samples (cg05771324: ∆β = +0.07; cg07585558: ∆β = +0.08) and four were hypomethylated in opioid-exposed samples (cg22393128: ∆β = −0.12; cg24415698: ∆β = −0.11; cg06547839: ∆β = −0.10; cg06347739: ∆β = −0.09). *PLD1* codes for Phospholipase D1, a phosphodiesterase involved in a range of metabolic and signaling pathways, including endocytosis and vesicular transport [38]. *RP11-250B2.3* and *RP11-32K4.1* both encode long intergenic non-coding (linc) RNAs, while *AL159152.1* codes for a microRNA (miRNA); each of these three genes has poorly-established function. *MGAM* codes for Maltase-Glucoamylase, a membrane digestive enzyme involved in starch digestion and linked to chronic diarrhea in children [39]. *ASCL2* codes for Achaete-Scute Family BHLH Transcription Factor 2, a protein involved in trophoblast proliferation in the placenta [40].

We identified 199 opioid exposure-associated DMRs that had a minimum smoothed FDR < 0.05. Of these, 94 (~47%) were hypomethylated, and 105 (~53%) were hypermethylated in opioid-exposed samples relative to controls. The top 15 DMRs (FDR < 0.05) associated with opioid exposure are shown in Table 2. One DMR on Chromosome 2, spanning 16 CpGs (1,322 bp), was mapped to the first exon of the Boule Homolog, RNA Binding Protein gene (*BOLL*; Figure 5A), which is involved in meiosis and gamete development [41]. In opioid-exposed samples, methylation levels were 9.0% higher on average at CpG sites within this region relative to control samples (min smoothed FDR = 8.5 × 10^−9^). Another DMR (1,120 bp) on Chromosome 10 spanning 10 CpGs was mapped to the first exon of the Potassium Calcium-Activated Channel Subfamily M Alpha 1 gene (*KCNMA1*; Figure 5B), which has been associated with substance use disorders [42,43]. Opioid exposed samples had methylation levels that were an average of 11.8% lower than control samples across this region (min smoothed FDR = 2.6 × 10^−5^).

### 3.3. Functional Enrichment

The 684 DMPs with both *p* ≤ 0.005 and |∆β| ≥ 0.05 were mapped to 258 annotated genes that were then used as input for enrichment analysis. The top 15 Gene Ontology (GO) enrichment terms (all unadjusted *p* ≤ 6.4 × 10^−4^ and all adjusted *p* ≤ 0.054) are shown in Figure 6A. Ten of the top fifteen GO enrichment terms met the adjusted threshold for significance, including regulation of synapse assembly (GO:0051963; adjusted *p* = 1.9 × 10^−4^; 8 genes), integral component of plasma membrane (GO:0005887; adjusted *p* = 6.8 × 10^−4^; 40 genes); positive regulation of nervous system development (GO:0051962; adjusted *p* = 0.010; 6 genes); positive regulation of synapse assembly (GO:0051965; adjusted *p* = 0.010; 5 genes), chemical synaptic transmission (GO:0007268; adjusted *p* = 0.010; 14 genes), presynapse assembly (GO:0099054; adjusted *p* = 0.010; 4 genes), modulation of chemical synaptic transmission (GO:0050804; adjusted *p* = 0.010; 8 genes), neuron differentiation (GO:0030182; adjusted *p* = 0.010; 10 genes), positive regulation of cell junction assembly (GO:1901890; adjusted *p* = 0.010; 6 genes), and generation of neurons (GO:0048699; adjusted *p* = 0.040; 10 genes). The top Kyoto Encyclopedia of Genes and Genomes (KEGG) enrichment terms are shown in Figure 6B. None of the KEGG enrichment terms reached the adjusted significance threshold.

We additionally performed enrichment analysis for the DMRs. A total of 199 DMRs (minimum smoothed FDR < 0.05) were mapped to 174 genes that were used as input. Of the top 15 GO enrichment terms and the top 9 KEGG enrichment terms (Figure 7), only one reached the adjusted threshold for significance: the GO term ‘integral component of plasma membrane’ (GO:0005887; adjusted *p* = 3.7 × 10^−4^; 31 genes). This term was also implicated in the DMP enrichment analysis. None of the KEGG enrichment terms reached the adjusted significance threshold.

## 4. Discussion

This is the first genome-wide study of placental DNA methylation alterations that were associated with opioid exposure during pregnancy. Although no CpG sites reached the adjusted significance threshold (*p* < 9.4 × 10^−8^), a subset of 684 CpG sites and affiliated genes with methylation influenced by in utero opioid exposure were identified using nominal thresholds (*p* ≤ 0.005 and |∆β| ≥ 0.05). Several of the top DMPs were annotated to genes of potential interest, including *PLD1*, a gene implicated in a variety of processes, including receptor endocytosis [44] and cytoskeletal organization [45]. The other isoform of this gene, *PLD2*, is required for μ-opioid receptor endocytosis and opioid desensitization [46]. The effects of aberrant *PLD1* regulation in the placenta are not clear, but as this gene is involved in a variety of processes for cellular homeostasis, this alteration could have a significant impact on placental function with subsequent effects on fetal development. We additionally identified a DMP in *MGAM,* a membrane digestive enzyme involved in starch digestion and linked to chronic diarrhea in children [39]. *MGAM* has been most thoroughly studied in the mucosa [47] and intestinal tissue [48], but considering the prevalence of feeding problems and low birth weight in opioid-exposed neonates [6,49], it is possible that dysregulation of this gene impacts metabolic processes in the placenta, impacting nutrient transfer to the developing fetus. One of the 684 DMPs used in the enrichment analysis mapped to the first exon of *OPRM1* (cg15085086; *p* = 0.004). In opioid-exposed samples, this site had 5.54% less methylation than controls. As we previously established an association between *OPRM1* and NOWS outcomes [25], it is likely that a larger study would be more successful in the identification of additional CpG sites annotated to *OPRM1* in association with opioid exposure.

We additionally identified several interesting genes annotated to DMRs that were associated with opioid exposure, including *KCNMA1* (Figure 5B). The KCNMA1 subunit is part of a large-conductance potassium channel that affects neuronal excitability and has been implicated in opioid analgesia [50], neuropathic pain [51], and kappa opioid receptor-mediated cardiomyocyte response to ischemia [52]. It has also been implicated in other substance use disorders (SUDs), including alcohol use disorder (AUD) [53,54] and methamphetamine use disorder (MUD) [55]. Additionally, *KCNMA1* has been found to have decreased gene expression in endometrial stromal fibroblasts during decidualization, supporting its function as a regulator of embryonic development [56]. The exact role of *KCNMA1* in the placenta, particularly in the context of opioid exposure, remains unclear. Future studies can better ascertain whether this is a viable target gene in the treatment and study of NOWS.

The results of our enrichment analyses for the top DMPs and DMRs indicate a disruption in gene pathways related to synaptic assembly and organization, chemical signal transduction, neuron differentiation, and nervous system development. There is substantial evidence from rodent models that prenatal opioid exposure influences neuron differentiation, synaptogenesis, and postsynaptic density protein expression in the central nervous system (CNS) [57,58,59]. While it is not feasible to analyze fetal or infant brain samples, the implication of these pathways in our placental enrichment analyses may be somewhat predictive of epigenetic adaptations present in the CNS [60].

This study has several limitations. First, it is limited by the small sample size that was underpowered to detect differences in DNA methylation at individual CpG sites that passed the strict FDR threshold for significance. Moreover, given that these were samples from subjects who did not sign informed consent, we were limited by few clinical variables (as well as missing data) to correlate with the samples. Future studies should include more detailed clinical and drug exposure information through a consented study-specific questionnaire. Second, accounting for additional variables in the linear regression, including maternal age, gestational age, and smoking status, could significantly improve the ability to identify significant associations between opioid exposure and placental DNA methylation. Third, there was also variability in opioid exposure type, which primarily included methadone (~53%), buprenorphine (~16%), and fentanyl (26%). Methadone and buprenorphine both have much longer elimination half-lives (~30 h) [61,62] versus that of fentanyl (3–7 h, i.v.) [63]. Future studies of larger sample sizes would better serve to determine the effects of individual opioid medications on placental DNA methylation. Fourth, beyond the epigenetic modifications induced by exposure to opioids themselves, there are likely additional implications related to the presentation of spontaneous opioid withdrawal during pregnancy. Future studies could better disentangle the effects of opioid agonism versus opioid withdrawal on placental DNA methylation and subsequent NOWS outcomes. Fifth, we need to further explore the function of those identified hypermethylated or hypomethylated CpG sites to clarify whether altered methylation at these CpG sites could influence their gene transcription. Finally, the villous placental tissue samples evaluated for this analysis contain a heterogeneous mix of cell populations, so our overall results do not give exact information on cell-specific DNA methylation changes. Ongoing studies evaluating gene methylation differences within isolated key placental cell populations will provide further insight into how gene expression changes can be correlated with altered placental function and fetal development.

## 5. Conclusions

This pilot work is the first step to the further examination of epigenetic variation in placental tissue within opioid-exposed pregnancies, with potential importance for improving neonatal outcomes and informing transgenerational risk. Future studies could include additional assays of epigenetic modifications (such as ATAC-seq and quantification of histone acetylation) for a more comprehensive understanding of how opioids influence epigenetic profiles and chromatin accessibility in the placenta. Furthermore, the inclusion of high-throughput RNA-seq would enable a concomitant analysis of transcriptomic and epigenetic adaptations and help facilitate the identification of genomic regions underlying variation in behavioral features of NOWS. Genes determined to be associated with NOWS severity and/or developmental outcomes could be potential targets for therapeutics [64,65,66].

## Figures and Tables

**Figure 1 biomedicines-10-01150-f001:**
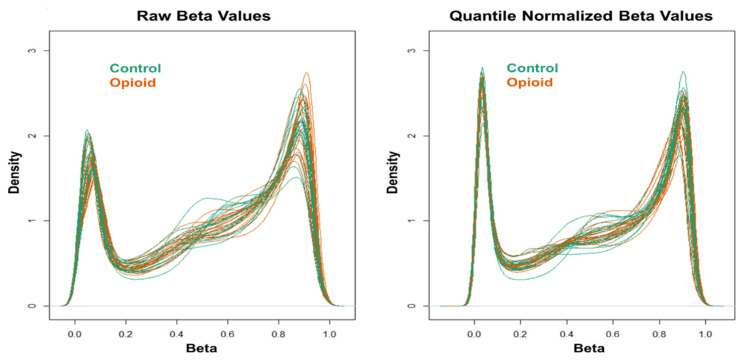
Density plots of raw (**left**) and quantile-normalized (**right**) beta values from opioid-exposed (*n* = 19) and control (*n* = 20) samples.

**Figure 2 biomedicines-10-01150-f002:**
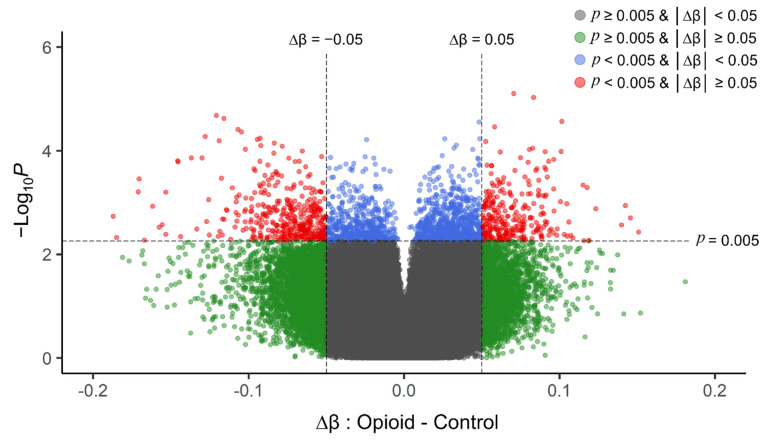
Volcano plot of unadjusted *p*-values (*y*-axis; −Log_10_ scale) versus delta beta values (opioid-exposed − controls; *x*-axis) for each CpG site in the filtered analysis (710,952 probes). Positive delta beta values indicate higher average methylation in opioid-exposed samples versus controls (and negative delta beta values indicate lower average methylation in opioid-exposed samples versus controls). The vertical dashed line represents an absolute delta beta cut-off of |∆β| ≥ 0.05 (green dots), the horizontal dashed line represents an unadjusted *p*-value cut-off of *p* < 0.005 (blue dots), and sites that met both cut-offs are shown in red. These top sites (red) were used as the input for subsequent enrichment analyses. The grey dots represent those CpG sites with |∆β| < 0.05 and *p* > 0.005. *p*-values were obtained from differential methylation analysis in limma (effect of opioid with fetal sex, fetal body weight, and batch as covariates).

**Figure 3 biomedicines-10-01150-f003:**
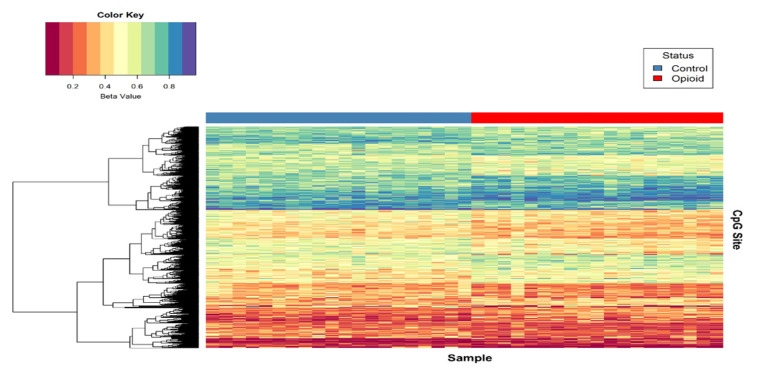
Heat map of normalized beta values for the top differentially methylated CpG sites associated with opioid treatment. Beta values for CpG sites with an unadjusted *p* < 0.005 and |∆β| ≥ 0.05 are shown (*n* = 684). Each row contains the beta value for a CpG site, color-coded from 0-1 (see color key, top left), and columns pertain to individual samples (*n* = 39). Samples were grouped by opioid-exposure status (top blue bar, left = controls; top red bar, right = opioid-exposed). Hierarchical sorting was performed by CpG site (row), and the dendrogram (left) indicates similarities in methylation trends across all CpG sites.

**Figure 4 biomedicines-10-01150-f004:**
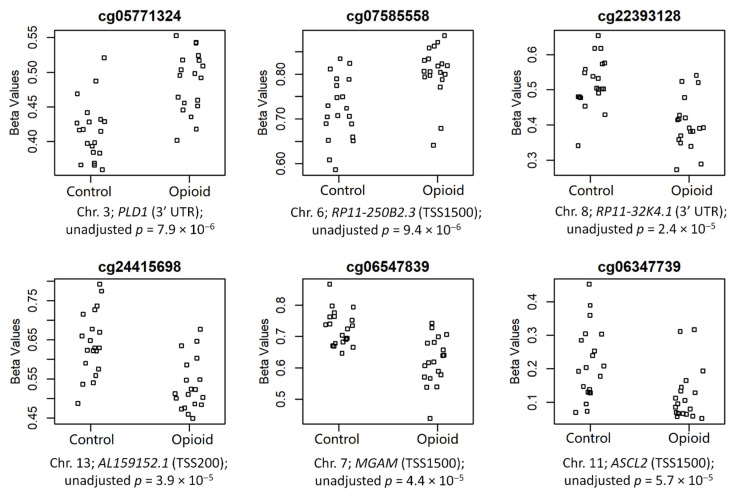
Normalized beta values are shown by groups (controls vs. opioid-exposed) for the top 6 annotated CpG sites associated with opioid exposure. CpG site names are above each plot, and chromosome numbers, associated genes, genomic regions, and unadjusted *p*-value are below each plot.

**Figure 5 biomedicines-10-01150-f005:**
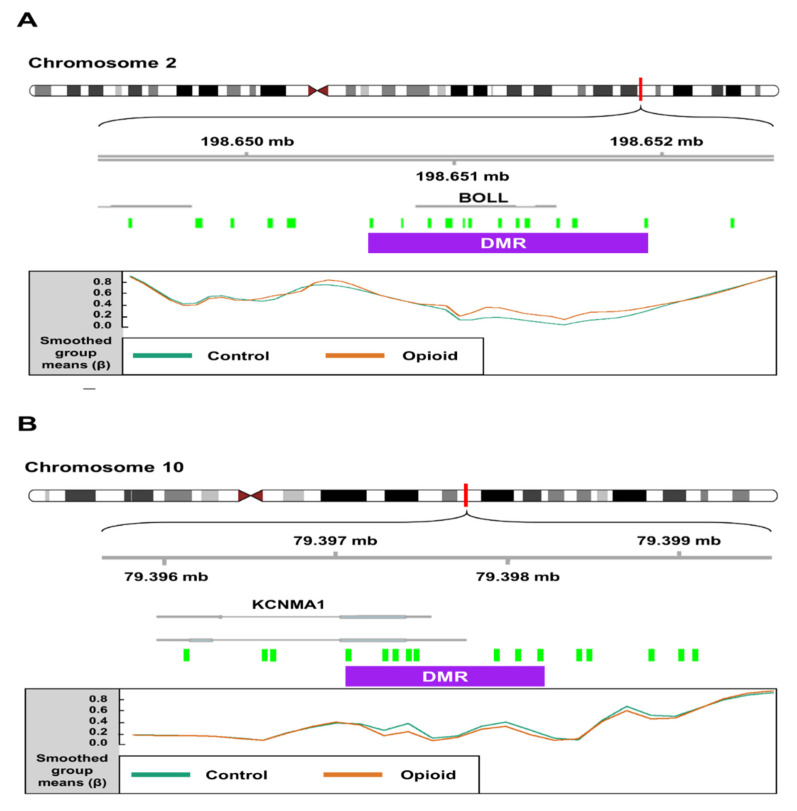
Schematics depicting differentially methylated regions (DMRs) associated with opioid exposure. (**A**) A DMR (1,322 bp; represented by purple box) on chromosome 2 spanning 16 CpG probes (green boxes) mapped to the first exon of the Boule Homolog, RNA Binding Protein gene (*BOLL*). In opioid-exposed samples, methylation levels were 9.02% higher on average at CpG sites within this region relative to control samples (min smoothed FDR = 8.5 × 10^−9^). (**B**) A DMR (1120 bp) on chromosome 10 spanning 10 CpG probes mapped to the first exon of the Potassium Calcium-Activated Channel Subfamily M Alpha 1 gene (*KCNMA1*). Opioid exposed samples had methylation levels that were an average of 11.8% lower than control samples across this region (min smoothed FDR = 2.6 × 10^−5^). Mean normalized beta values across the DMR are shown for each group (control vs. opioid exposed) at the bottom of each panel.

**Figure 6 biomedicines-10-01150-f006:**
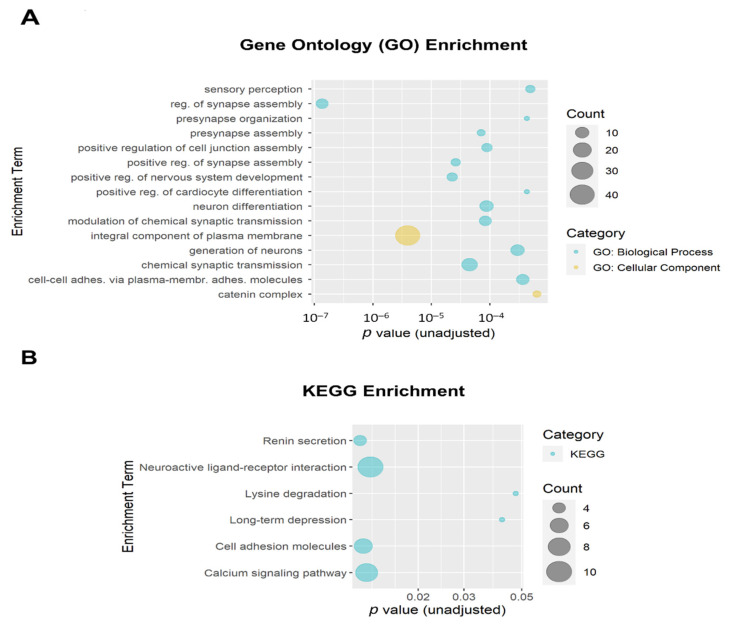
Bubble plots of gene enrichment results from the top CpG sites (unadjusted *p* < 0.005 and |∆β| ≥ 0.05; mapped to 258 genes) associated with opioid exposure. (**A**) The top 15 Gene Ontology (GO) enrichment terms. GO terms are color-coded by subcategory (Biological Process vs. Cellular Component). All unadjusted *p* ≤ 6.4 × 10^−4^ and all adjusted *p* ≤ 0.054. Ten GO enrichment terms met the adjusted threshold for significance (*p* < 0.05). (**B**) The top 6 Kyoto Encyclopedia of Genes and Genomes (KEGG) enrichment terms. The size of the bubble for each enrichment term corresponds to the number of enriched genes within that pathway. None of the KEGG enrichment terms reached the adjusted significance threshold.

**Figure 7 biomedicines-10-01150-f007:**
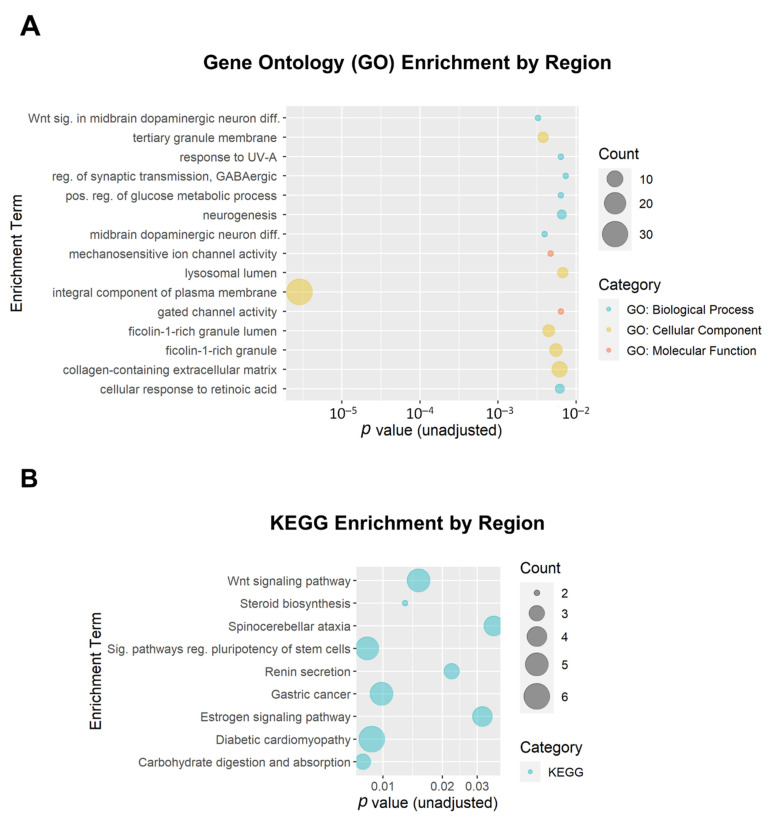
Bubble plots of gene enrichment results from the top differentially methylated regions associated with opioid exposure. A total of 199 DMRs (minimum smoothed FDR < 0.05) mapped to 174 genes that were used as input for the enrichment analysis. (**A**) The top 15 Gene Ontology (GO) enrichment terms. GO terms are color-coded by subcategory (Biological Process, Molecular Function, or Cellular Component). All unadjusted *p* ≤ 0.007. (**B**) The top 9 Kyoto Encyclopedia of Genes and Genomes (KEGG) enrichment terms. The size of the bubble for each enrichment term corresponds to the number of enriched genes within that pathway.

**Table 1 biomedicines-10-01150-t001:** Maternal and infant demographics.

Demographics	Opioid (*n* = 19)	Control (*n* = 20)	*p*-Value
Maternal age in years, mean (SD)	28.8 (5.1)	32.7 (3.4)	0.010
Missing, *n*	3	2	
Maternal Ethnicity/Race, *n* (%)	Non-Hispanic	Non-Hispanic	
White	18 (94.7%)	19 (95%)	0.970
Black	1 (5.3%)	1 (5%)	
Maternal Opioid, *n* (%)		N/A	N/A
Methadone	10 (52.6%)		
Buprenorphine	3 (15.8%)		
Other prescription opioid	1 (5.3%)		
Unprescribed fentanyl	5 (26.3%)		
Gestational age at delivery (weeks), median (IQR)	39.5 (36.6–40.1)	39.6 (38.5–40.5)	0.220
Missing, *n* (%)	3 (15.8%)	2 (10%)	
Cesarean section delivery, *n* (%)	6 (37.5%)	6 (33.3%)	1.000
Missing, *n* (%)	3 (15.8%)	2 (10%)	
Infant sex, *n* (%)			
Female	10 (52.6%)	8 (40%)	0.44
Male	9 (47.4%)	12 (60%)	
Birthweight (g), median (IQR)	3061 (2825–3462)	3401 (3065–3795)	0.05

SD = standard deviation. IQR = inter-quartile range. N/A = Not applicable. There were no opioid-exposed samples in the control group.

**Table 2 biomedicines-10-01150-t002:** Top 15 differentially methylated regions (DMRs) annotated in opioid-exposed samples by minimum smoothed FDR (|∆β| ≥ 0.05).

Chr. #	Width (bp)	# CpGs	Min Smoothed FDR	Mean Diff. (β)	Overlapping Genes
chr2	1477	12	2.4 × 10^−14^	−0.085	*ANKRD53*, *AC007040.11*
chr2	1308	12	5.0 × 10^−11^	0.059	*C2orf70*
chr15	548	7	1.6 × 10^−10^	0.065	*TSPAN3*
chr11	1136	14	1.5 × 10^−9^	0.098	*MSANTD4*
chr2	1322	16	8.5 × 10^−9^	0.09	*BOLL*
chr8	530	9	4.2 × 10^−6^	−0.061	*FDFT1*
chr8	1164	10	1.2 × 10^−5^	−0.109	*ZNF572*
chr16	1144	16	2.1 × 10^−5^	0.056	*CYBA*
chr10	1120	10	2.6 × 10^−5^	−0.118	*KCNMA1*
chr16	1396	8	2.6 × 10^−5^	0.058	*IRX3*
chr3	822	5	3.2 × 10^−5^	0.054	*MIR4792*
chr3	999	11	3.3 × 10^−5^	−0.056	*HLTF-AS1, HLTF*
chr19	978	7	4.8 × 10^−5^	−0.122	*B3GNT3*
chr7	1063	8	8.0 × 10^−5^	−0.079	*WNT2*
chr12	469	2	1.2 × 10^−4^	−0.081	*MGAM*

## Data Availability

The data that support the findings of this study are available from the corresponding author upon reasonable request.

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
