# Peer review of "Effect of Prenatal Opioid Exposure on the Human Placental Methylome"

_biomedicines, 2022, doi:10.3390/biomedicines10051150_

Round 1

Reviewer 1 Report

This is an interesting pilot study providing important insight into the potential epigenetic changes that are present at parturition in the placenta from mothers who have a history of opioid use disorder relative to controls mothers lacking a history of opioid misuse. The authors provide a good background indicating that the placenta is an established site of epigenetic modifications with subsequent differences in gene expression and tissue differentiation. The goal of the current pilot study was to perform a methylome-wide evaluation of placental DNA methylation changes after prenatal opioid exposure, as measured at the time of birth. Placental DNA methylomes were profiled (using the Illumina Infinium HumanMethylationEPIC BeadChip), and differentially methylated CpG sites associated with opioid exposure were identified with a linear model (‘limma’ R package), and lastly, the function of genes was further annotated by bioinformatics analysis with Gene Ontology and Kyoto Encyclopedia of Genes and Genomes pathway enrichment analyses of differentially methylated genes. The presentation of the analyses are clear and straightforward. The data are unique providing a necessary step forward to the field in terms of gaining insight into the role of placenta in impacting fetal development. Only a few recommended improvements in the manuscript are needed.

Abstract

The concluding sentence in the abstract is a bit odd. There should be a prior sentence introducing the concept of NOWS and speculated impact on CNS structure and function, as the preceeding sentence states that observed epigenetic changes are related to neuronal synaptic structure and communication.

Introduction

While it is understood that the manuscript is not focused on detailed descriptions of molecular biology, some mention of the function of methylated CpG should be included to provide context; for example, that CpG islands occur more frequently in the promotor region and that methylation stabilizes gene silencing.

Methods

  1. The rationale for collecting the maternal side vs. the fetal side of the placenta is needed.
  2. “Samples were dropped off if more…” page 4, line 149, 150 and 154 – please modify phraseology to reduce jargon.

Discussion

  1. There are differences in the known action of opioids at the mu-opioid receptor and other receptors, for example methadone buprenorphine, vs fentanyl. The impact of these different opioids acting on the opioid receptors as well as receptors in addition to the opioid receptors (e.g. methadone acts at the NMDA receptor) is an important consideration.
  2. Implications for hypomethylated vs. hypermethylated CpGs should be considered.

Reviewer 2 Report

Kristyn N. Borrelli et al investigated the epigenetic effects of addictive drugs in placenta using

 the Illumina Infinium Human Methylation EPIC BeadChip, a robust approach with extensive coverage of CpG islands, genes, and enhancers. The results showed methylation changes of genes involved in nervous system. Although this is a pilot study, the overall results suggest that prenatal opioid exposure may induce specific epigenetic changes in the placenta.

Placental collection: The authors report: 'a sample of villous placental tissue was collected from the maternal side of the placenta'. The placenta is very heterogeneous and composed of different cell types/structures. They should specify whether the decidua (placental component of maternal origin) has been removed, as inappropriate sampling of the placenta (such as not removing the decidua), may be responsible for undesirable variations. Generally, placental samples from studies aimed at highlighting environmental effects on pregnancy are collected from the fetal side to avoid maternal contamination. The authors should explain the choice of sampling the maternal side of the placenta.

Methylation analyses

The section on Illumina EPIC DNA methylation array assay and the raw data processing (section 2.4.3) is well written and thorough, especially the filtering process. However, it would be more comprehensive to specify the number of sites interrogated throughout the genome and how many of them remain after the filtering procedures. For example, when the authors describe why some probes were excluded from the analysis (e.g., cross-reactive probes, probes that overlap with genetic variants at targeted CpG sites, probes with genetic variants that overlap with the probe body, and those mapped to X and Y chromosomes), the number of sites excluded from the analysis and the number that are considered should be specified.

Patient race classification (table 1)The reasons that race/ethnicity were assessed in the study should be described (e.g., in the Methods section). In general, ethnicity is preferable to race.
